# Online Convex Optimization with Unconstrained Domains and Losses

**Ashok Cutkosky**
Department of Computer Science
Stanford University
ashokc@cs.stanford.edu

**Kwabena Boahen**
Department of Bioengineering
Stanford University
boahen@stanford.edu

## Abstract

We propose an online convex optimization algorithm (RESCALEDEXP) that achieves optimal regret in the unconstrained setting without prior knowledge of any bounds on the loss functions. We prove a lower bound showing an *exponential* separation between the regret of existing algorithms that require a known bound on the loss functions and any algorithm that does not require such knowledge. RESCALEDEXP matches this lower bound asymptotically in the number of iterations. RESCALEDEXP is naturally hyperparameter-free and we demonstrate empirically that it matches prior optimization algorithms that require hyperparameter optimization.

## 1 Online Convex Optimization

Online Convex Optimization (OCO) [1, 2] provides an elegant framework for modeling noisy, antagonistic or changing environments. The problem can be stated formally with the help of the following definitions:

**Convex Set:** A set $W$ is convex if $W$ is contained in some real vector space and $tw + (1-t)w' \in W$ for all $w, w' \in W$ and $t \in [0, 1]$.

**Convex Function:** $f : W \to \mathbb{R}$ is a convex function if $f(tw + (1-t)w') \le tf(w) + (1-t)f(w')$ for all $w, w' \in W$ and $t \in [0, 1]$.

An OCO problem is a game of repeated rounds in which on round $t$ a learner first chooses an element $w_t$ in some convex space $W$, then receives a convex loss function $\ell_t$, and suffers loss $\ell_t(w_t)$. The *regret* of the learner with respect to some other $u \in W$ is defined by

$$R_T(u) = \sum_{t=1}^{T} \ell_t(w_t) - \ell_t(u)$$

The objective is to design an algorithm that can achieve low regret with respect to any $u$, even in the face of adversarially chosen $\ell_t$.

Many practical problems can be formulated as OCO problems. For example, the stochastic optimization problems found widely throughout machine learning have exactly the same form, but with i.i.d. loss functions, a subset of the OCO problems. In this setting the goal is to identify a vector $w_\star$ with low generalization error ($\mathbb{E}[\ell(w_\star) - \ell(u)]$). We can solve this by running an OCO algorithm for $T$ rounds and setting $w_\star$ to be the average value of $w_t$. By online-to-batch conversion results [3, 4], the generalization error is bounded by the expectation of the regret over the $\ell_t$ divided by $T$. Thus, OCO algorithms can be used to solve stochastic optimization problems while also performing well in non-i.i.d. settings.

The regret of an OCO problem is upper-bounded by the regret on a corresponding Online Linear Optimization (OLO) problem, in which each $\ell_t$ is further constrained to be a *linear* function: $\ell_t(w) = g_t \cdot w_t$ for some $g_t$. The reduction follows, with the help of one more definition:

**Subgradient:** $g \in W$ is a subgradient of $f$ at $w$, denoted $g \in \partial f(w)$, if and only if $f(w) + g \cdot (w' - w) \leq f(w')$ for all $w'$. Note that $\partial f(w) \neq \emptyset$ if $f$ is convex.[1]

To reduce OCO to OLO, suppose $g_t \in \partial \ell_t(w_t)$, and consider replacing $\ell_t(w)$ with the linear approximation $g_t \cdot w$. Then using the definition of subgradient,

$$R_T(u) = \sum_{t=1}^{T} \ell_t(w_t) - \ell_t(u) \leq \sum_{t=1}^{T} g_t(w_t - u) = \sum_{t=1}^{T} g_t w_t - g_t u$$

so that replacing $\ell_t(w)$ with $g_t \cdot w$ can only make the problem more difficult. All of the analysis in this paper therefore addresses OLO, accessing convex losses functions only through subgradients.

There are two major factors that influence the regret of OLO algorithms: the size of the space $W$ and the size of the subgradients $g_t$. When $W$ is a bounded set (the "constrained" case), then given $B = \max_{w \in W} \|w\|$, there exist OLO algorithms [5, 6] that can achieve $R_T(u) \leq O\left(B L_{\max} \sqrt{T}\right)$ without knowing $L_{\max} = \max_t \|g_t\|$. When $W$ is unbounded (the "unconstrained" case), then given $L_{\max}$, there exist algorithms [7, 8, 9] that achieve $R_T(u) \leq \tilde{O}(\|u\| \log(\|u\|) L_{\max} \sqrt{T})$ or $R_t(u) \leq \tilde{O}(\|u\| \sqrt{\log(\|u\|)} L_{\max} \sqrt{T})$, where $\tilde{O}$ hides factors that depend logarithmically on $L_{\max}$ and $T$. These algorithms are known to be optimal (up to constants) for their respective regimes [10, 7]. All algorithms for the unconstrained setting to-date require knowledge of $L_{\max}$ to achieve these optimal bounds.[2] Thus a natural question is: *can we achieve $O(\|u\| \log(\|u\|))$ regret in the unconstrained, unknown-$L_{\max}$ setting?* This problem has been posed as a COLT 2016 open problem [12], and is solved in this paper.

A simple approach is to maintain an estimate of $L_{\max}$ and double it whenever we see a new $g_t$ that violates the assumed bound (the so-called "doubling trick"), thereby turning a known-$L_{\max}$ algorithm into an unknown-$L_{\max}$ algorithm. This strategy fails for previous known-$L_{\max}$ algorithms because their analysis makes strong use of the assumption that each and every $\|g_t\|$ is bounded by $L_{\max}$. The existence of even a small number of bound-violating $g_t$ can throw off the entire analysis.

In this paper, we prove that it is actually impossible to achieve regret $O\left(\|u\| \log(\|u\|) L_{\max} \sqrt{T} + L_{\max} \exp\left[\left(\max_t \frac{\|g_t\|}{L(t)}\right)^{1/2-\epsilon}\right]\right)$ for any $\epsilon > 0$ where $L_{\max}$ and $L(t) = \max_{t' < t} \|g_{t'}\|$ are unknown in advance (Section 2). This immediately rules out the "ideal" bound of $\tilde{O}(\|u\| \sqrt{\log(\|u\|)} L_{\max} \sqrt{T})$ which is possible in the known-$L_{\max}$ case. Secondly, we provide an algorithm, RESCALEDEXP, that matches our lower bound without prior knowledge of $L_{\max}$, leading to a naturally hyperparameter-free algorithm (Section 3). To our knowledge, this is the first algorithm to address the unknown-$L_{\max}$ issue while maintaining $O(\|u\| \log \|u\|)$ dependence on $u$. Finally, we present empirical results showing that RESCALEDEXP performs well in practice (Section 4).

## 2   Lower Bound with Unknown $L_{\max}$

The following theorem rules out algorithms that achieve regret $O(u \log(u) L_{\max} \sqrt{T})$ without prior knowledge of $L_{\max}$. In fact, any such algorithm must pay an up-front penalty that is *exponential* in $T$. This lower bound resolves a COLT 2016 open problem (Parameter-Free and Scale-Free Online Algorithms) [12] in the negative.

**Theorem 1.** *For any constants $c, k, \epsilon > 0$, there exists a $T$ and an adversarial strategy picking $g_t \in \mathbb{R}$ in response to $w_t \in \mathbb{R}$ such that regret is:*

$$R_T(u) = \sum_{t=1}^{T} g_t w_t - g_t u$$

$$\geq (k + c\|u\|\log\|u\|)L_{\max}\sqrt{T}\log(L_{\max}+1) + kL_{\max}\exp((2T)^{1/2-\epsilon})$$

$$\geq (k + c\|u\|\log\|u\|)L_{\max}\sqrt{T}\log(L_{\max}+1) + kL_{\max}\exp\left[\left(\max_t \frac{\|g_t\|}{L(t)}\right)^{1/2-\epsilon}\right]$$

*for some $u \in \mathbb{R}$ where $L_{\max} = \max_{t \leq T}\|g_t\|$ and $L(t) = \max_{t' < t}\|g_{t'}\|$.*

*Proof.* We prove the theorem by showing that for sufficiently large $T$, the adversary can "checkmate" the learner by presenting it only with the subgradient $g_t = -1$. If the learner fails to have $w_t$ increase quickly, then there is a $u \gg 1$ against which the learner has high regret. On the other hand, if the learner ever does make $w_t$ higher than a particular threshold, the adversary immediately punishes the learner with a subgradient $g_t = 2T$, again resulting in high regret.

Let $T$ be large enough such that both of the following hold:

$$\frac{T}{4}\exp(\tfrac{T^{1/2}}{4\log(2)c}) > k\log(2)\sqrt{T} + k\exp((2T)^{1/2-\epsilon}) \tag{1}$$

$$\frac{T}{2}\exp(\tfrac{T^{1/2}}{4\log(2)c}) > 2kT\exp((2T)^{1/2-\epsilon}) + 2kT\sqrt{T}\log(2T+1) \tag{2}$$

The adversary plays the following strategy: for all $t \leq T$, so long as $w_t < \frac{1}{2}\exp(T^{1/2}/4\log(2)c)$, give $g_t = -1$. As soon as $w_t \geq \frac{1}{2}\exp(T^{1/2}/4\log(2)c)$, give $g_t = 2T$ and $g_t = 0$ for all subsequent $t$. Let's analyze the regret at time $T$ in these two cases.

**Case 1:** $w_t < \frac{1}{2}\exp(T^{1/2}/4\log(2)c)$ **for all** $t$:

In this case, let $u = \exp(T^{1/2}/4\log(2)c)$. Then $L_{\max} = 1$, $\max_t \frac{\|g_t\|}{L(t)} = 1$, and using (1) the learner's regret is at least

$$R_T(u) \geq Tu - T\frac{1}{2}\exp(\tfrac{T^{1/2}}{4\log(2)c})$$

$$= \tfrac{1}{2}Tu$$

$$= cu\log(u)\sqrt{T}\log(2) + \tfrac{T}{4}\exp(\tfrac{T^{1/2}}{4\log(2)c})$$

$$> cu\log(u)L_{\max}\sqrt{T}\log(L_{\max}+1) + kL_{\max}\sqrt{T}\log(L_{\max}+1) + kL_{\max}\exp((2T)^{1/2-\epsilon})$$

$$= (k + cu\log u)L_{\max}\sqrt{T}\log(L_{\max}+1) + kL_{\max}\exp\left[\max_t(2T)^{1/2-\epsilon}\right]$$

**Case 2:** $w_t \geq \frac{1}{2}\exp(T^{1/2}/4\log(2)c)$ **for some** $t$:

In this case, $L_{\max} = 2T$ and $\max_t \frac{\|g_t\|}{L(t)} = 2T$. For $u = 0$, using (2), the regret is at least

$$R_T(u) \geq \tfrac{T}{2}\exp(\tfrac{T^{1/2}}{4\log(2)c})$$

$$\geq 2kT\exp((2T)^{1/2-\epsilon}) + 2kT\sqrt{T}\log(2T+1)$$

$$= kL_{\max}\exp((2T)^{1/2-\epsilon}) + kL_{\max}\sqrt{T}\log(L_{\max}+1)$$

$$= (k + cu\log u)L_{\max}\sqrt{T}\log(L_{\max}+1) + kL_{\max}\exp\left[\max_t(2T)^{1/2-\epsilon}\right]$$

$$\square$$

The exponential lower-bound arises because the learner has to move exponentially fast in order to deal with exponentially far away $u$, but then experiences exponential regret if the adversary provides a gradient of unprecedented magnitude in the opposite direction. However, if we play against an adversary that is constrained to give loss vectors $\|g_t\| \leq L_{\max}$ for some $L_{\max}$ that does not grow with time, or if the losses do not grow too quickly, then we can still achieve $R_T(u) = O(\|u\|\log(\|u\|)L_{\max}\sqrt{T})$ asymptotically without knowing $L_{\max}$. In the following sections we describe an algorithm that accomplishes this.

## 3 RESCALEDEXP

Our algorithm, RESCALEDEXP, adapts to the unknown $L_{\max}$ using a guess-and-double strategy that is robust to a small number of bound-violating $g_t$s. We initialize a guess $L$ for $L_{\max}$ to $\|g_1\|$. Then we run a novel known-$L_{\max}$ algorithm that can achieve good regret in the unconstrained $u$ setting. As soon as we see a $g_t$ with $\|g_t\| > 2L$, we update our guess to $\|g_t\|$ and restart the known-$L_{\max}$ algorithm. To prove that this scheme is effective, we show (Lemma 3) that our known-$L_{\max}$ algorithm does not suffer too much regret when it sees a $g_t$ that violates its assumed bound.

Our known-$L_{\max}$ algorithm uses the Follow-the-Regularized-Leader (FTRL) framework. FTRL is an intuitive way to design OCO algorithms [13]: Given functions $\psi_t : W \to \mathbb{R}$, at time $T$ we play $w_T = \operatorname{argmin} \left[ \psi_{T-1}(w) + \sum_{t=1}^{T-1} \ell_t(w) \right]$. The functions $\psi_t$ are called regularizers. A large number of OCO algorithms (e.g. gradient descent) can be cleanly formulated as instances of this framework.

Our known-$L_{\max}$ algorithm is FTRL with regularizers $\psi_t(w) = \psi(w)/\eta_t$, where $\psi(w) = (\|w\| + 1) \log(\|w\| + 1) - \|w\|$ and $\eta_t$ is a scale-factor that we adapt over time. Specifically, we set $\eta_t^{-1} = k\sqrt{2}\sqrt{M_t + \|g\|_{1:t}^2}$, where we use the compressed sum notations $g_{1:T} = \sum_{t=1}^{T} g_t$ and $\|g\|_{1:T}^2 = \sum_{t=1}^{T} \|g_t\|^2$. $M_t$ is defined recursively by $M_0 = 0$ and $M_t = \max(M_{t-1}, \|g_{1:t}\|/p - \|g\|_{1:t}^2)$, so that $M_t \geq M_{t-1}$, and $M_t + \|g\|_{1:t}^2 \geq \|g_{1:t}\|/p$. $k$ and $p$ are constants: $k = \sqrt{2}$ and $p = L_{\max}^{-1}$.

RESCALEDEXP's strategy is to maintain an estimate $L_t$ of $L_{\max}$ at all time steps. Whenever it observes $\|g_t\| \geq 2L_t$, it updates $L_{t+1} = \|g_t\|$. We call periods during which $L_t$ is constant *epochs*. Every time it updates $L_t$, it restarts our known-$L_{\max}$ algorithm with $p = \frac{1}{L_t}$, beginning a new epoch. Notice that since $L_t$ at least doubles every epoch, there will be at most $\log_2(L_{\max}/L_1) + 1$ total epochs. To address edge cases, we set $w_t = 0$ until we suffer a non-constant loss function, and we set the initial value of $L_t$ to be the first non-zero $g_t$. Pseudo-code is given in Algorithm 1, and Theorem 2 states our regret bound. For simplicity, we re-index so that that $g_1$ is the first non-zero gradient received. No regret is suffered when $g_t = 0$ so this does not affect our analysis.

---

**Algorithm 1** RESCALEDEXP

> **Initialize:** $k \leftarrow \sqrt{2}$, $M_0 \leftarrow 0$, $w_1 \leftarrow 0$, $t_\star \leftarrow 1$ // $t_\star$ is the start-time of the current epoch.
> **for** $t = 1$ **to** $T$ **do**
>    Play $w_t$, receive subgradient $g_t \in \partial \ell_t(w_t)$.
>    **if** $t = 1$ **then**
>       $L_1 \leftarrow \|g_1\|$
>       $p \leftarrow 1/L_1$
>    **end if**
>    $M_t \leftarrow \max(M_{t-1}, \|g_{t_\star:t}\|/p - \|g\|_{t_\star:t}^2)$.
>    $\eta_t \leftarrow \frac{1}{k\sqrt{2(M_t + \|g\|_{t_\star:t}^2)}}$
>    //Set $w_{t+1}$ using FTRL update
>    $w_{t+1} \leftarrow -\frac{g_{t_\star:t}}{\|g_{t_\star:t}\|} \left[\exp(\eta_t \|g_{t_\star:t}\|) - 1\right]$ // $= \operatorname{argmin}_w \left[ \frac{\psi(w)}{\eta_t} + g_{t_\star:t} w \right]$
>    **if** $\|g_t\| > 2L_t$ **then**
>       //Begin a new epoch: update $L$ and restart FTRL
>       $L_{t+1} \leftarrow \|g_t\|$
>       $p \leftarrow 1/L_{t+1}$
>       $t_\star \leftarrow t + 1$
>       $M_t \leftarrow 0$
>       $w_{t+1} \leftarrow 0$
>    **else**
>       $L_{t+1} \leftarrow L_t$
>    **end if**
> **end for**

---

**Theorem 2.** *Let $W$ be a separable real inner-product space with corresponding norm $\| \cdot \|$ and suppose (with mild abuse of notation) every loss function $\ell_t : W \to \mathbb{R}$ has some subgradient $g_t \in W^*$ at $w_t$ such that $g_t(w) = g_t \cdot w$ for some $g_t \in W$. Let $M_{\max} = \max_t M_t$. Then if $L_{\max} = \max_t \|g_t\|$*

*and* $L(t) = \max_{t' < t} \|g_t\|$, rescaledexp *achieves regret:*

$$R_T(u) \le (2\psi(u) + 96) \left( \log_2 \left( \frac{L_{\max}}{L_1} \right) + 1 \right) \sqrt{M_{\max} + \|g\|_{1:T}^2}$$

$$+ 8L_{\max} \left( \log_2 \left( \frac{L_{\max}}{L_1} \right) + 1 \right) \min \left[ \exp \left( 8 \max_t \frac{\|g_t\|^2}{L(t)^2} \right), \exp(\sqrt{T/2}) \right]$$

$$= O \left( L_{\max} \log \left( \frac{L_{\max}}{L_1} \right) \left[ (\|u\| \log(\|u\|) + 2)\sqrt{T} + \exp \left( 8 \max_t \frac{\|g_t\|^2}{L(t)^2} \right) \right] \right)$$

The conditions on $W$ in Theorem 2 are fairly mild. In particular they are satisfied whenever $W$ is finite-dimensional and in most kernel method settings [14]. In the kernel method setting, $W$ is an RKHS of functions $\mathcal{X} \to \mathbb{R}$ and our losses take the form $\ell_t(w) = \ell_t(\langle w, k_{x_t} \rangle)$ where $k_{x_t}$ is the representing element in $W$ of some $x_t \in \mathcal{X}$, so that $g_t = \bar{g}_t k_{x_t}$ where $\bar{g}_t \in \partial \ell_t(\langle w, k_{x_t} \rangle)$.

Although we nearly match our lower-bound exponential term of $\exp((2T)^{1/2-\epsilon})$, in order to have a practical algorithm we need to do much better. Fortunately, the $\max_t \frac{\|g_t\|^2}{L(t)^2}$ term may be significantly smaller when the losses are not fully adversarial. For example, if the loss vectors $g_t$ satisfy $\|g_t\| = t^2$, then the exponential term in our bound reduces to a manageable constant even though $\|g_t\|$ is growing quickly without bound.

To prove Theorem 2, we bound the regret of RESCALEDEXP during each epoch. Recall that during an epoch, RESCALEDEXP is running FTRL with $\psi_t(w) = \psi(w)/\eta_t$. Therefore our first order of business is to analyze the regret of FTRL across one of these epochs, which we do in Lemma 3 (proved in appendix):

**Lemma 3.** *Set* $k = \sqrt{2}$. *Suppose* $\|g_t\| \le L$ *for* $t < T$, $1/L \le p \le 2/L$, $g_T \le L_{\max}$ *and* $L_{\max} \ge L$. *Let* $W_{\max} = \max_{t \in [1,T]} \|w_t\|$. *Then the regret of FTRL with regularizers* $\psi_t(w) = \psi(w)/\eta_t$ *is:*

$$R_T(u) \le \psi(u)/\eta_T + 96\sqrt{M_T + \|g\|_{1:T}^2} + 2L_{\max} \min \left[ W_{\max}, 4 \exp \left( 4 \frac{L_{\max}^2}{L^2} \right), \exp(\sqrt{T/2}) \right]$$

$$\le (2\psi(u) + 96) \sqrt{\sum_{t=1}^{T-1} L|g_t| + L_{\max}^2} + 8L_{\max} \min \left[ \exp \left( \frac{4L_{\max}^2}{L^2} \right), \exp(\sqrt{T/2}) \right]$$

$$\le L_{\max}(2((\|u\| + 1)\log(\|u\| + 1) - \|u\|) + 96)\sqrt{T} + 8L_{\max} \min \left[ e^{\frac{4L_{\max}^2}{L^2}}, e^{\sqrt{T/2}} \right]$$

Lemma 3 requires us to know the value of $L$ in order to set $p$. However, the crucial point is that it encompasses the case in which $L$ is misspecified on the last loss vector. This allows us to show that RESCALEDEXP does not suffer too much by updating $p$ on-the-fly.

*Proof of Theorem 2.* The theorem follows by applying Lemma 3 to each epoch in which $L_t$ is constant.

Let $1 = t_1, t_2, t_3, \cdots, t_n$ be the various increasing values of $t_\star$ (as defined in Algorithm 1), and we define $t_{n+1} = T + 1$. Then define

$$R_{a:b}(u) = \sum_{t=a}^{b-1} g_t(w_t - u)$$

so that $R_T(u) \le \sum_{j=1}^n R_{t_j:t_{j+1}}(u)$. We will bound $R_{t_j:t_{j+1}}(u)$ for each $j$.

Fix a particular $j < n$. Then $R_{t_j:t_{j+1}}(u)$ is simply the regret of FTRL with $k = \sqrt{2}$, $p = \frac{1}{L_{t_j}}$, $\eta_t = \frac{1}{k\sqrt{2(M_t + \|g\|_{t_j:t}^2)}}$ and regularizers $\psi(w)/\eta_t$. By definition of $L_t$, for $t \in [1, t_{j+1} - 2]$ we have $\|g_t\| \le 2L_{t_j}$. Further, if $L = \max_{t \in [1, t_{j+1} - 2]} \|g_t\|$ we have $L \ge L_{t_j}$. Therefore, $L_{t_j} \le L \le 2L_{t_j}$ so that $\frac{1}{L} \le p \le \frac{2}{L}$. Further, we have $\|g_{t_{j+1}-1}\|/L_{t_j} \le 2 \max_t \|g_t\|/L(t)$. Thus by Lemma 3 we

have

$$R_{t_j:t_{j+1}}(u) \leq \psi(u)/\eta_{t_{j+1}-1} + 96\sqrt{M_{t_{j+1}-1} + \|g\|_{t_j:t_{j+1}-1}^2}$$

$$+ 2L_{\max} \min\left[W_{\max}, 4\exp\left(4\frac{\|g_{t_{j+1}-1}\|^2}{L_{t_j}^2}\right), \exp\left(\frac{\sqrt{t_{j+1}-t_j}}{\sqrt{2}}\right)\right]$$

$$\leq \psi(u)/\eta_{t_{j+1}-1} + 96\sqrt{M_{\max} + \|g\|_{t_j:t_{j+1}-1}^2} + 8L_{\max} \min\left[e^{8\max_t \frac{\|g_t\|^2}{L(t)^2}}, e^{\sqrt{T/2}}\right]$$

$$\leq (2\psi(u)+96)\sqrt{M_{\max} + \|g\|_{1:T}^2} + 8L_{\max} \min\left[\exp\left(8\max_t \frac{\|g_t\|^2}{L(t)^2}\right), \exp(\sqrt{T/2})\right]$$

Summing across epochs, we have

$$R_T(u) = \sum_{j=1}^n R_{t_j:t_{j+1}}(u)$$

$$\leq n\left[(2\psi(u)+96)\sqrt{M_{\max} + \|g\|_{1:T}^2} + 8L_{\max} \min\left[\exp\left(8\max_t \frac{\|g_t\|^2}{L(t)^2}\right), \exp\left(\sqrt{T/2}\right)\right]\right]$$

Observe that $n \leq \log_2(L_{\max}/L_1) + 1$ to prove the first line of the theorem. The big-Oh expression follows from the inequality: $M_{t_{j+1}-1} \leq L_{t_j}\sum_{t=t_j}^{t_{j+1}-1}\|g_t\| \leq L_{\max}\sum_{t=1}^T \|g_t\|$. □

Our specific choices for $k$ and $p$ are somewhat arbitrary. We suspect (although we do not prove) that the preceding theorems are true for larger values of $k$ and any $p$ inversely proportional to $L_t$, albeit with differing constants. In Section 4 we perform experiments using the values for $k$, $p$ and $L_t$ described in Algorithm 1. In keeping with the spirit of designing a hyperparameter-free algorithm, no attempt was made to empirically optimize these values at any time.

## 4 Experiments

### 4.1 Linear Classification

To validate our theoretical results in practice, we evaluated RESCALEDEXP on 8 classification datasets. The data for each task was pulled from the libsvm website [15], and can be found individually in a variety of sources [16, 17, 18, 19, 20, 21, 22]. We use linear classifiers with hinge-loss for each task and we compare RESCALEDEXP to five other optimization algorithms: ADAGRAD [5], SCALEINVARIANT [23], PISTOL [24], ADAM [25], and ADADELTA [26]. Each of these algorithms requires tuning of some hyperparameter for unconstrained problems with unknown $L_{\max}$ (usually a scale-factor on a learning rate). In contrast, our RESCALEDEXP requires no such tuning.

We evaluate each algorithm with the average loss after one pass through the data, computing a prediction, an error, and an update to model parameters for each example in the dataset. Note that this is not the same as a cross-validated error, but is closer to the notion of regret addressed in our theorems. We plot this average loss versus hyperparameter setting for each dataset in Figures 1 and 2. These data bear out the effectiveness of RESCALEDEXP: while it is not unilaterally the highest performer on all datasets, it shows remarkable robustness across datasets with zero manual tuning.

### 4.2 Convolutional Neural Networks

We also evaluated RESCALEDEXP on two convolutional neural network models. These models have demonstrated remarkable success in computer vision tasks and are becoming increasingly more popular in a variety of areas, but can require significant hyperparameter tuning to train. We consider the MNIST [18] and CIFAR-10 [27] image classification tasks.

Our MNIST architecture consisted of two consecutive $5 \times 5$ convolution and $2 \times 2$ max-pooling layers followed by a 512-neuron fully-connected layer. Our CIFAR-10 architecture was two consecutive $5 \times 5$ convolution and $3 \times 3$ max-pooling layers followed by a 384-neuron fully-connected layer and a 192-neuron fully-connected layer.

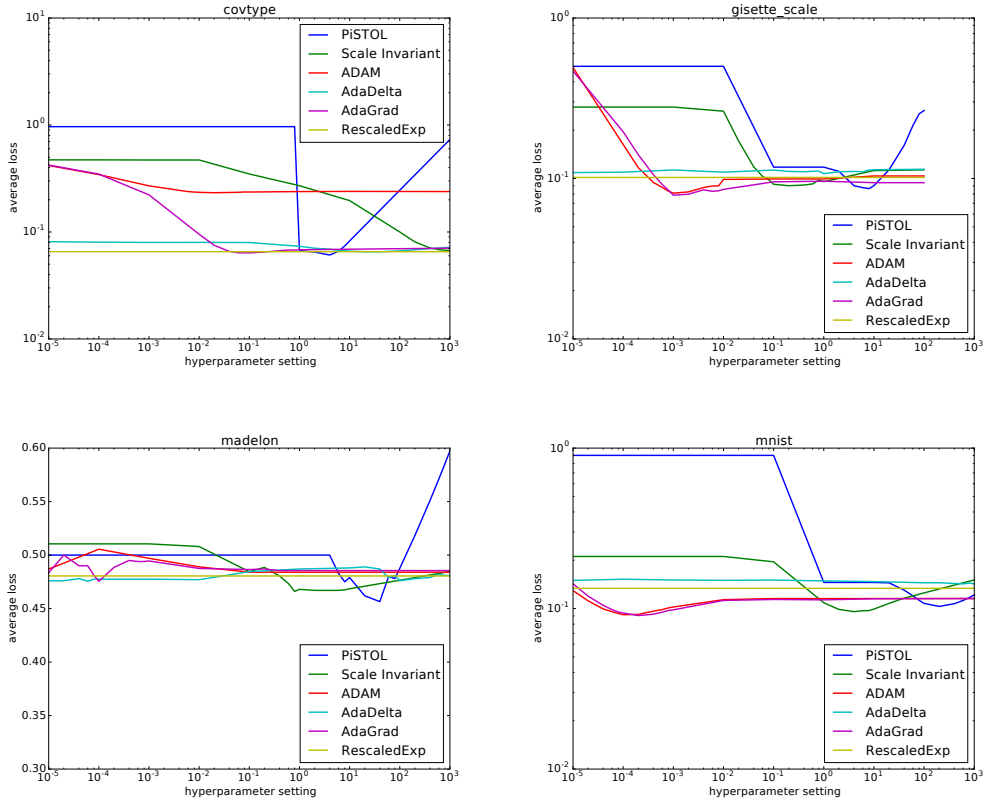

Figure 1: Average loss vs hyperparameter setting for each algorithm across each dataset. RESCALED-EXP has no hyperparameters and so is represented by a flat yellow line. Many of the other algorithms display large sensitivity to hyperparameter setting.

These models are highly non-convex, so that none of our theoretical analysis applies. Our use of RESCALEDEXP is motivated by the fact that in practice convex methods are used to train these models. We found that RESCALEDEXP can match the performance of other popular algorithms (see Figure 3).

In order to achieve this performance, we made a slight modification to RESCALEDEXP: when we update $L_t$, instead of resetting $w_t$ to zero, we re-center the algorithm about the previous prediction point. We provide no theoretical justification for this modification, but only note that it makes intuitive sense in stochastic optimization problems, where one can reasonably expect that the previous prediction vector is closer to the optimal value than zero.

## 5 Conclusions

We have presented RESCALEDEXP, an Online Convex Optimization algorithm that achieves regret $\tilde{O}(\|u\| \log(\|u\|) L_{\max} \sqrt{T} + \exp(8 \max_t \|g_t\|^2 / L(t)^2))$ where $L_{\max} = \max_t \|g_t\|$ is unknown in advance. Since RESCALEDEXP does not use any prior-knowledge about the losses or comparison vector $u$, it is hyperparameter free and so does not require any tuning of learning rates. We also prove a lower-bound showing that any algorithm that addresses the unknown-$L_{\max}$ scenario must suffer an exponential penalty in the regret. We compare RESCALEDEXP to prior optimization algorithms empirically and show that it matches their performance.

While our lower-bound matches our regret bound for RESCALEDEXP in terms of $T$, clearly there is much work to be done. For example, when RESCALEDEXP is run on the adversarial loss sequence presented in Theorem 1, its regret matches the lower-bound, suggesting that the optimality gap could be improved with superior analysis. We also hope that our lower-bound inspires work in algorithms that adapt to non-adversarial properties of the losses to avoid the exponential penalty.

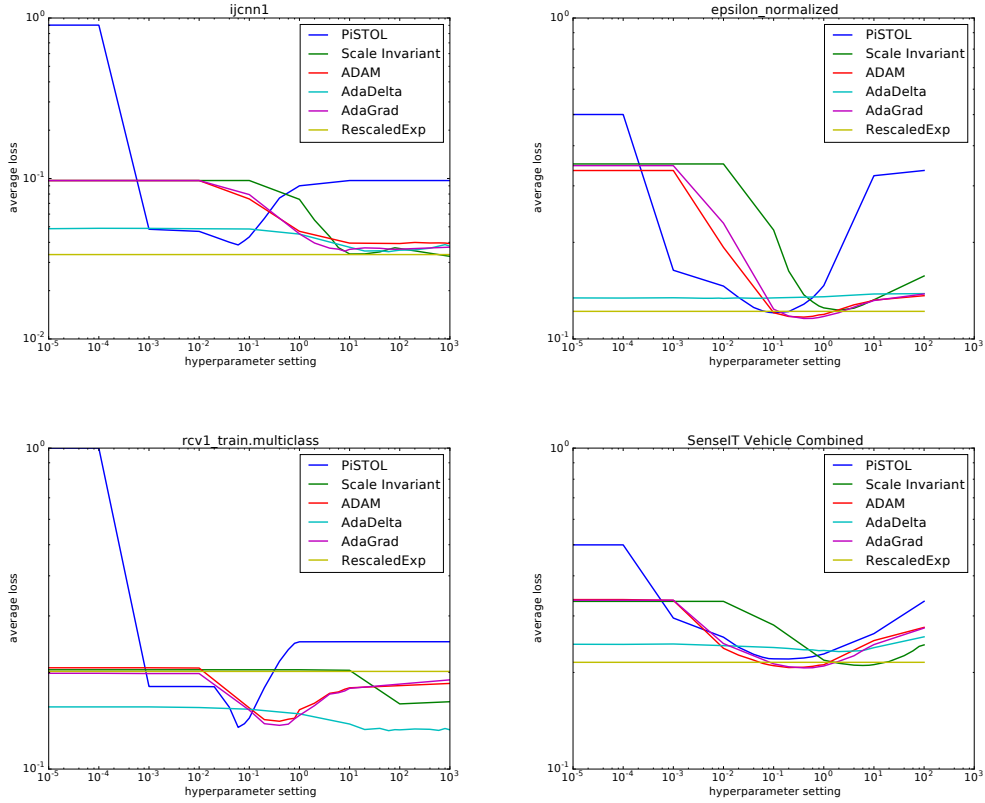

Figure 2: Average loss vs hyperparameter setting, continued from Figure 1.

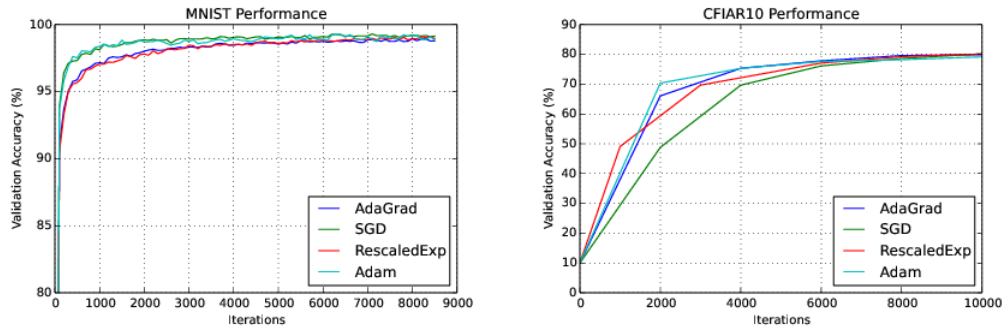

Figure 3: We compare RESCALEDEXP to ADAM, ADAGRAD, and stochastic gradient descent (SGD), with learning-rate hyperparameter optimization for the latter three algorithms. All algorithms achieve a final validation accuracy of 99% on MNIST and 84%, 84%, 83% and 85% respectively on CIFAR-10 (after 40000 iterations).

## Footnotes

[1]In full generality, a subgradient is an element of the dual space $W^*$. However, we will only consider cases where the subgradient is naturally identified with an element in the original space $W$ (e.g. $W$ is finite dimensional) so that the definition in terms of dot-products suffices.

[2]There are algorithms that do not require $L_{\max}$, but achieve only regret $O(\|u\|^2)$ [11]

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
