[Supplementary Material · unconstrainedOCOappendix.pdf]

# A  Follow-the-Regularized-Leader (FTRL) Regret

Recall that the FTRL algorithm uses the strategy $w_{t+1} = \text{argmin}\, \psi_t(w) + \sum_{t'=1}^{t} \ell_{t'}(w)$, where the functions $\psi_t$ are called regularizers.

**Theorem 4.** *FTRL with regularizers $\psi_t$ and $\psi_0(w_1) = 0$ obtains regret:*

$$R_t(u) \le \psi_T(u) + \sum_{t=1}^{T} \psi_{t-1}(w_{t+1}) - \psi_t(w_{t+1}) + \ell_t(w_t) - \ell_t(w_{t+1}) \tag{3}$$

*Further, if the losses are linear $\ell_t(w) = g_t \cdot w$ and $\psi_t(w) = \frac{1}{\eta_t}\psi(w)$ for some values $\eta_t$ and fixed function $\psi$, then the regret is*

$$R_t(u) \le \frac{1}{\eta_T}\psi(u) + \sum_{t=1}^{T} \left( \frac{1}{\eta_{t-1}} - \frac{1}{\eta_t} \right) \psi(w_{t+1}) + g_t \cdot (w_t - w_{t+1}) \tag{4}$$

*Proof.* The first part follows from some algebraic manipulations:

$$\sum_{t=1}^{T} \ell_t(u) + \psi_T(u) \ge \psi_T(w_{T+1}) + \sum_{t=1}^{T} \ell_t(w_{T+1})$$

$$-\sum_{t=1}^{T} \ell_t(u) \le \psi_T(u) - \psi_T(w_{T+1}) - \sum_{t=1}^{T} \ell_t(w_{T+1})$$

$$
\begin{aligned}
R_T(u) &= \sum_{t=1}^{T} \ell_t(w_t) - \sum_{t=1}^{T} \ell_t(u) \\
&\le \psi_T(u) - \psi_T(w_{T+1}) + \sum_{t=1}^{T} \ell_t(w_t) - \ell_t(w_{T+1}) \\
&= \psi_T(u) - \psi_T(w_{T+1}) + \ell_T(w_T) - \ell_T(w_{T+1}) + R_{T-1}(w_{T+1}) \\
&\le \psi_T(u) - \psi_T(w_{T+1}) + \ell_T(w_T) - \ell_T(w_{T+1}) + \\
&\quad + \sum_{t=1}^{T-1} \psi_t(w_{t+2}) - \psi_t(w_{t+1}) + \ell_t(w_t) - \ell_t(w_{t+1}) \\
&= \psi_T(u) + \ell_1(w_1) - \ell_1(w_2) - \psi_1(w_2) \\
&\quad + \sum_{t=2}^{T} \psi_{t-1}(w_{t+1}) - \psi_t(w_{t+1}) + \ell_t(w_t) - \ell_t(w_{t+1}) \\
&= \psi_T(u) + \sum_{t=1}^{T} \psi_{t-1}(w_{t+1}) - \psi_t(w_{t+1}) + \ell_t(w_t) - \ell_t(w_{t+1})
\end{aligned}
$$

where we're assuming $\psi_0(w_1) = 0$ in the last step.

Now let's specialize to the case of linear losses $\ell_t(w) = g_t \cdot w$ and regularizers of the form $\psi_t(w) = \frac{1}{\eta_t}\psi(w)$ for some fixed regularizer $\psi$ and varying scalings $\eta_t$. Plugging this into the previous bound gives:

$$R_t(u) \le \frac{1}{\eta_T}\psi(u) + \sum_{t=1}^{T} \left( \frac{1}{\eta_{t-1}} - \frac{1}{\eta_t} \right) \psi(w_{t+1}) + g_t \cdot (w_t - w_{t+1})$$

$\square$

While this formulation of the regret of FTRL is sufficient for our needs, our analysis is not tight. We refer the reader to [28] for a stronger FTRL bound that can improve constants in some analyses.

# B  Proof of Lemma 3

We start off by computing the FTRL updates with regularizers $\psi(w)/\eta_t$:

$$\nabla\psi(w) = \log(\|w\| + 1)\frac{w}{\|w\|}$$

so that

$$w_{T+1} = \operatorname{argmin} \frac{1}{\eta_T}\psi(w) + \sum_{t=1}^{T} g_t \cdot w$$
$$= -\frac{g_{1:t}}{\|g_{1:t}\|}(\exp(\eta_T\|g_{1:T}\|) - 1)$$

Our goal will be to show that the terms $\left(\frac{1}{\eta_{t-1}} - \frac{1}{\eta_t}\right)\psi(w_{t+1}) + g_t \cdot (w_t - w_{t+1})$ in the sum in (4) are negative. In particular, note that sequence of $\eta_t$ is non-increasing so that $\left(\frac{1}{\eta_{t-1}} - \frac{1}{\eta_t}\right)\psi(w_{t+1}) \le 0$ for all $t$. Thus our strategy will be to bound $g_t \cdot (w_t - w_{t+1})$.

## B.1   Reduction to one dimension

In order to bound $\left(\frac{1}{\eta_{t-1}} - \frac{1}{\eta_t}\right)\psi(w_{t+1}) + g_t \cdot (w_t - w_{t+1})$, we first show that it suffices to consider the case when $g_t$ and $g_{1:t-1}$ are co-linear.

**Theorem 5.** *Let $W$ be a separable inner-product space and suppose (with mild abuse of notation) every loss function $\ell_t : W \to \mathbb{R}$ has some subgradient $g_t \in W^*$ such that $g_t w = \langle g_t, w\rangle$ for some $g_t \in W$. Suppose we run an FTRL algorithm with regularizers $\frac{1}{\eta_t}\psi(\|w\|)$ on loss functions $\ell_t$ such that $w_{t+1} = \frac{g_{1:t}}{\|g_{1:t}\|}f(\eta_t\|g_{1:t}\|)$ for some function $f$ for all $t$ where $\eta_t = \frac{c}{\sqrt{M_t + \|g\|_{1:t}^2}}$ for some constant $c$. Then for any $g_t$ with $\|g_t\| = L$, both $(\eta_{t-1}^{-1} - \eta_t^{-1})\psi(\|w_{t+1}\|) + g_t(w_t - w_{t+1})$ and $g_t(w_t - w_{t+1})$ are maximized when $g_t$ is a scalar multiple of $g_{1:t-1}$.*

*Proof.* The proof is an application of Lagrange multipliers. Our Lagrangian for $(\eta_{t-1}^{-1} - \eta_t^{-1})\psi(\|w_{t+1}\|) + g_t(w_t - w_{t+1})$ is

$$\mathcal{L} = (\eta_{t-1}^{-1} - \eta_t^{-1})\psi(\|w_{t+1}\|) + g_t(w_t - w_{t+1}) + \lambda\|g_t\|^2/2$$
$$= (\eta_{t-1}^{-1} - \eta_t^{-1})\psi(f(\eta_t\|g_{1:t}\|)) + g_t\left(w_t - \frac{g_{1:t}}{\|g_{1:t}\|}f(\eta_t\|g_{1:t}\|)\right) + \lambda\frac{\|g_t\|^2}{2}$$

Fix a countable orthonormal basis of $W$. For a vector $v \in W$ we let $v_i$ be the projection of $v$ along the $i$th basis vector of our countable orthonormal basis. We denote the action of $\nabla\mathcal{L}$ on the $i$th basis vector by $\nabla\mathcal{L}_i$.

Then we have

$$\nabla\mathcal{L}_i = \lambda g_{t,i} + w_{t,i} - w_{t+1,i} - \frac{g_{t,i}}{\|g_{1:t}\|}f(\eta_t\|g_{1:t}\|)$$
$$+ \sum_j \frac{g_{t,j}(g_{1:t})_j}{\|g_{1:t}\|^3}(g_{1:t})_i f(\eta_t\|g_{1:t}\|)$$
$$- \sum_j \frac{(g_{1:t})_j g_{t,j}}{\|g_{1:t}\|}f'(\eta_t\|g_{1:t}\|)\left[\frac{(g_{1:t})_i\eta_t}{\|g_{1:t}\|} - \frac{\|g_{1:t}\|c\left(\frac{\partial M_t}{\partial g_{t,i}} + 2g_{t,i}\right)}{2(M_t + \|g\|_{1:t}^2)^{3/2}}\right]$$
$$+ (\eta_{t-1}^{-1} - \eta_t^{-1})\psi'(f(\eta_t\|g_{1:t}\|))f'(\eta_t\|g_{1:t}\|)\left[\frac{(g_{1:t})_i\eta_t}{\|g_{1:t}\|} - \frac{\|g_{1:t}\|c\left(\frac{\partial M_t}{\partial g_{t,i}} + 2g_{t,i}\right)}{2(M_t + \|g\|_{1:t}^2)^{3/2}}\right]$$
$$- \psi(f(\eta_t\|g_{1:t}\|))\frac{\frac{\partial M_t}{\partial g_{t,i}} + 2g_{t,i}}{2c\sqrt{M_t + \|g\|_{1:t}^2}}$$
$$= \lambda g_{t,i} + w_{t,i} - w_{t+1,i} + Ag_{t,i} + B(g_{1:t-1})_i + C\frac{\partial M_t}{\partial g_{t,i}}$$

where $A$, $B$ and $C$ do not depend on $i$. Since $w_{t,i}$ and $w_{t+1,i}$ are scalar multiples of $g_{1:t-1}$ and $g_{1:t}$ respectively, we can reassign the variables $A$ and $B$ to write

$$\nabla\mathcal{L}_i = Ag_{t,i} + B(g_{1:t-1})_i + C\frac{\partial M_t}{\partial g_{t,i}}$$

Now we compute

$$\frac{\partial M_t}{\partial g_{t,i}} = \frac{\partial \max(M_{t-1}, \|g_{1:t}\|/p - \|g\|_{1:t}^2)}{\partial g_{t,i}}$$

$$= \begin{cases} 0 & : M_t = M_{t-1} \\ \frac{(g_{1:t})_i}{p\|g_{1:t}\|} - 2g_{t,i} & : M_t \neq M_{t-1} \end{cases}$$

Thus after again reassigning the variables $A$ and $B$ we have

$$\nabla \mathcal{L}_i = A g_{t,i} + B(g_{1:t-1})_i$$

Therefore we can only have $\nabla \mathcal{L} = 0$ if $g_t$ is a scalar multiple of $g_{1:t-1}$ as desired.

For $g_t(w_t - w_{t+1})$, we apply exactly the same argument. The Lagrangian is

$$\mathcal{L} = g_t(w_t - w_{t+1}) + \lambda \|g_t\|^2/2$$

$$= g_t \left( w_t - \frac{g_{1:t}}{\|g_{1:t}\|} f(\eta_t \|g_{1:t}\|) \right) + \lambda \frac{\|g_t\|^2}{2}$$

and differentiating we have

$$\nabla \mathcal{L}_i = \lambda g_{t,i} + w_{t,i} - w_{t+1,i} - \frac{g_{t,i}}{\|g_{1:t}\|} f(\eta_t \|g_{1:t}\|)$$

$$+ \sum_j \frac{g_{t,j}(g_{1:t})_j}{\|g_{1:t}\|^3} (g_{1:t})_i f(\eta_t \|g_{1:t}\|)$$

$$- \sum_j \frac{(g_{1:t})_j g_{t,j}}{\|g_{1:t}\|} f'(\eta_t \|g_{1:t}\|) \left[ \frac{(g_{1:t})_i \eta_t}{\|g_{1:t}\|} - \frac{\|g_{1:t}\| c \left( \frac{\partial M_t}{\partial g_{t,i}} + 2g_{t,i} \right)}{2(M_t + \|g\|_{1:t}^2)^{3/2}} \right]$$

$$= \lambda g_{t,i} + w_{t,i} - w_{t+1,i} + A g_{t,i} + B(g_{1:t-1})_i + C \frac{\partial M_t}{\partial g_{t,i}}$$

$$= A g_{t,i} + B(g_{1:t-1})_i$$

so that again we are done.

$\square$

We make the following intuitive definition:

**Definition 6.** *For any vector $v \in W$, define $sign(v) = \frac{v}{\|v\|}$.*

In the next section, we prove bounds on the quantity $(\eta_{t-1}^{-1} - \eta_t^{-1})\psi(\|w_{t+1}\|) + g_t(w_t - w_{t+1})$. By Theorem 5 this quantity is maximized when $sign(g_t) = \pm sign(g_{1:t-1})$ and so we consider only this case.

## B.2 One dimensional FTRL

In this section we analyze the regret of our FTRL algorithm with the end-goal of proving Lemma 3. We make heavy use of Theorem 5 to allow us to consider only the case $sign(g_t) = \pm sign(g_{1:t-1})$. In this setting we may identify the 1-dimensional space spanned by $g_t$ and $g_{1:t-1}$ with $\mathbb{R}$. Thus whenever we are operating under the assumption $sign(g_t) = sign(g_{1:t-1})$ we will use $|\cdot|$ in place of $\|\cdot\|$ and occasionally assume $g_{1:t-1} > 0$ as this holds WLOG. We feel that this notation and assumption aids intuition in visualizing the following results.

**Lemma 7.** *Suppose $sign(g_t) = sign(g_{1:t-1})$. Then*

$$|\eta_{t-1}\|g_{1:t-1}\| - \eta_t\|g_{1:t}\|| \leq \eta_t \|g_t\| \tag{5}$$

*Suppose instead that $sign(g_t) = -sign(g_{1:t-1})$ and also $\|g_t\| \leq L$. Then we still have:*

$$|\eta_{t-1}\|g_{1:t-1}\| - \eta_t\|g_{1:t}\|| \leq \left( 1 + \frac{pL}{2} \right) \eta_t \|g_t\| \tag{6}$$

*Proof.* First, suppose $sign(g_t) = sign(g_{1:t-1})$. Then $sign(g_{1:t}) = sign(g_{1:t-1})$. WLOG, assume $g_{1:t-1} > 0$. Notice that $\eta_t g_{1:t}$ is an increasing function of $g_t$ for $g_t > 0$ because $\eta_t g_{1:t}$ is proportional to either $g_{1:t}$ or $\sqrt{g_{1:t}}$ depending on whether $M_t = M_{t-1}$ or not. Then since $\eta_t < \eta_{t-1}$ we have

$$|\eta_{t-1}g_{1:t-1} - \eta_t g_{1:t}| = \eta_t g_{1:t} - \eta_{t-1}g_{1:t-1}$$

$$\leq \eta_t g_{1:t} - \eta_t g_{1:t-1}$$

$$= \eta_t |g_t|$$

so that (5) holds.

Now suppose $\text{sign}(g_t) = -\text{sign}(g_{1:t-1})$ and $\|g_t\| \leq L$. We consider two cases.

**Case 1:** $\eta_t|g_{1:t}| \geq \eta_{t-1}|g_{1:t-1}|$:

Since $\eta_{t-1} \geq \eta_t$, we have

$$\eta_t|g_{1:t}| \geq \eta_{t-1}|g_{1:t-1}|$$
$$\eta_t|g_{1:t}| \geq \eta_t|g_{1:t-1}|$$
$$|g_{1:t}| \geq |g_{1:t-1}|$$
$$|g_t| \geq |g_{1:t}|$$

where the last line follows since $\text{sign}(g_{1:t-1}) = -\text{sign}(g_t)$. Therefore:

$$|\eta_{t-1}|g_{1:t-1}| - \eta_t|g_{1:t}|| \leq \eta_t|g_{1:t}| \leq \eta_t|g_t|$$

so that we are done.

**Case 2:** $\eta_t|g_{1:t}| \leq \eta_{t-1}|g_{1:t-1}|$:

When $g_t < -g_{1:t-1}$ and $\eta_t|g_{1:t}| \leq \eta_{t-1}|g_{1:t-1}|$, $|\eta_{t-1}|g_{1:t-1}| - \eta_t|g_{1:t}||$ is a decreasing function of $|g_t|$ because $\eta_t|g_{t:1}|$ is an increasing function of $|g_t|$ for $g_t < -g_{1:t-1}$. Therefore it suffices to consider the case $g_t \geq -g_{1:t-1}$, so that $\text{sign}(g_{1:t}) = \text{sign}(g_{1:t-1})$ and $|g_{1:t}| \leq |g_{1:t-1}|$:

Since $|g_{1:t}| \leq |g_{1:t-1}|$, we have $M_t = M_{t-1}$ so that we can write:

$$\eta_{t-1}g_{1:t-1} - \eta_t g_{1:t} = -g_t\eta_t + g_{1:t-1}(\eta_{t-1} - \eta_t)$$

$$= |g_t|\eta_t + g_{1:t-1}\left(\frac{1}{k\sqrt{2}\sqrt{M_{t-1} + \|g\|_{1:t-1}^2}} - \frac{1}{k\sqrt{2}\sqrt{M_t + \|g\|_{1:t-1}^2 + g_t^2}}\right)$$

$$= |g_t|\eta_t + \frac{g_{1:t-1}}{k\sqrt{2}}\left(\frac{1}{\sqrt{M_{t-1} + \|g\|_{1:t-1}^2}} - \frac{1}{\sqrt{M_{t-1} + \|g\|_{1:t-1}^2 + g_t^2}}\right)$$

$$\leq |g_t|\eta_t + \frac{g_{1:t-1}}{k\sqrt{2}\sqrt{M_t + \|g\|_{1:t-1}^2 + g_t^2}}\left(\frac{\sqrt{M_{t-1} + \|g\|_{1:t-1}^2 + g_t^2}}{\sqrt{M_{t-1} + \|g\|_{1:t-1}^2}} - 1\right)$$

$$\leq |g_t|\eta_t + g_{1:t-1}\eta_t\left(1 + \frac{g_t^2}{2(M_{t-1} + \|g\|_{1:t-1}^2)} - 1\right)$$

$$\leq |g_t|\eta_t + \eta_t\frac{g_{1:t-1}g_t^2}{2(M_{t-1} + \|g\|_{1:t-1}^2)}$$

$$\leq |g_t|\eta_t(1 + \frac{pL}{2})$$

we have used the identity $\sqrt{X + g_t^2} \leq \sqrt{X} + \frac{g_t^2}{2\sqrt{X}}$ between lines 4 and 5, and the last line follows because $|g_t| \leq L$ and $M_{t-1} + \|g\|_{1:t-1}^2 \geq |g_{1:t-1}|/p$. □

**Lemma 8.** *If*

$$\|w_T\| \geq \exp\left(\frac{\sqrt{pB}}{k\sqrt{2}}\right) - 1$$

*then*

$$\|g_{1:T-1}\| \geq B$$

*Proof.* First note that by definition of $M_{T-1}$ and $\eta_{T-1}$, $\eta_{T-1}\|g_{1:T-1}\| \leq \frac{\sqrt{p\|g_{1:T-1}\|}}{k\sqrt{2}}$. The proof now follows from some algebra:

$$\exp\left(\frac{\sqrt{pB}}{k\sqrt{2}}\right) \leq \|w_T\| + 1$$

$$= \exp(\eta_{T-1}\|g_{1:T-1}\|)$$

$$\leq \exp\left(\frac{\sqrt{p\|g_{1:T-1}\|}}{k\sqrt{2}}\right)$$

Taking squares of logs and rearranging now gives the desired inequality. □

We have the following immediate corollary:

**Corollary 9.** *Suppose* $sign(g_t) = \pm sign(g_{1:t-1})$, $\|g_t\| \leq L$, *and*

$$\|w_t\| \geq \exp\left(\frac{\sqrt{pL}}{k\sqrt{2}}\right) - 1$$

*Then* $sign(g_{1:t}) = sign(g_{1:t-1})$.

Now we begin analysis of the sum term in (4).

**Lemma 10.** *Suppose* $sign(g_{1:t}) = sign(g_{1:t-1})$ *and* $|g_t| \leq L$. *Then*

$$|w_t - w_{t+1}| \leq |g_t|\eta_t(|w_{t+1}| + 1)\left(1 + \frac{pL}{2}\right)\exp\left[g_t\eta_t\left(1 + \frac{pL}{2}\right)\right]$$

*Proof.* Since $sign(g_{1:t}) = sign(g_{1:t-1})$, we have:

$$\begin{aligned}
|w_t - w_{t+1}| &= |sign(g_{1:t-1})\left[\exp\left(\eta_{t-1}|g_{1:t-1}|\right) - 1\right] - [sign(g_{1:t})\exp\left(\eta_t|g_{1:t}|\right) - 1]| \\
&= |\exp\left(\eta_{t-1}|g_{1:t-1}|\right) - \exp\left(\eta_t|g_{1:t}|\right)| \\
&= (|w_{t+1}| + 1)|\exp\left(\eta_{t-1}|g_{1:t-1}| - \eta_t|g_{1:t}|\right) - 1|
\end{aligned}$$

where the last line uses the definition of $w_{t+1}$ to observe that $|w_{t+1}| + 1 = \exp(\eta_t|g_{1:t}|)$. Now we consider two cases: either $\eta_{t-1}|g_{1:t-1}| < \eta_t|g_{1:t}|$ or not.

**Case 1:** $\eta_{t-1}|g_{1:t-1}| < \eta_t|g_{1:t}|$:

By convexity of exp, we have

$$\begin{aligned}
|w_t - w_{t+1}| &\leq (|w_{t+1}| + 1)|\exp\left(\eta_{t-1}|g_{1:t-1}| - \eta_t|g_{1:t}|\right) - 1| \\
&\leq (|w_{t+1}| + 1)|\eta_{t-1}|g_{1:t-1}| - \eta_t|g_{1:t}|| \\
&\leq (|w_{t+1}| + 1)\left(1 + \frac{pL}{2}\right)\eta_t|g_t|
\end{aligned}$$

so that the lemma holds.

**Case 2:** $\eta_{t-1}|g_{1:t-1}| \geq \eta_t|g_{1:t}|$:

Again by convexity of exp we have

$$\begin{aligned}
|w_t - w_{t+1}| &\leq (|w_{t+1}| + 1)|\exp\left(\eta_{t-1}|g_{1:t-1}| - \eta_t|g_{1:t}|\right) - 1| \\
&\leq (|w_{t+1}| + 1)|\eta_{t-1}|g_{1:t-1}| - \eta_t|g_{1:t}||\exp\left(\eta_{t-1}|g_{1:t-1}| - \eta_t|g_{1:t}|\right) \\
&\leq (|w_{t+1}| + 1)\left(1 + \frac{pL}{2}\right)\exp\left[\eta_t|g_t|\left(1 + \frac{pL}{2}\right)\right]\eta_t|g_t|
\end{aligned}$$

so that the lemma still holds. $\square$

The next lemma is the main workhorse of our regret bounds:

**Lemma 11.** *Suppose* $\|g_t\| \leq L$ *and either of the following holds:*

1. *$p \leq \frac{2}{L}$, $k = \sqrt{2}$, and $\|w_t\| \geq 15$.*

2. *$k = \sqrt{2}$, $pL \geq 1$, and $\|w_t\| \geq 4\exp(p^2L^2)$.*

*Then*

$$\left(\frac{1}{\eta_{t-1}} - \frac{1}{\eta_t}\right)\psi(w_{t+1}) + g_t(w_t - w_{t+1}) \leq 0 \tag{7}$$

*Further, inequality (7) holds for any $k$ and sufficiently large $L$ if $\|w_t\| \geq \exp((pL)^2)$.*

*Proof.* By Theorem 5 it suffices to consider the case $sign(g_t) = \pm sign(g_{1:t-1})$, so that we may adopt our identification with $\mathbb{R}$ and use of $|\cdot|$ throughout this proof.

For $p \leq \frac{2}{L}$, $k = \sqrt{2}$ we have $15 > \exp(\frac{\sqrt{pL}}{k\sqrt{2}}) - 1$ and for sufficiently large $L$, $\exp((pL)^2) > \exp(\frac{\sqrt{pL}}{k\sqrt{2}}) - 1$. Therefore in all cases $|w_t| \geq \exp(\frac{\sqrt{pL}}{k\sqrt{2}}) - 1$ so that by Corollary 9 and Lemma 10 we have

$$g_t \cdot (w_t - w_{t+1}) \leq \eta_t g_t^2(|w_{t+1}| + 1)\left(1 + \frac{pL}{2}\right)\exp\left[\eta_t g_t\left(1 + \frac{pL}{2}\right)\right] \tag{8}$$

First, we prove that (7) is guaranteed if the following holds:

$$|w_{t+1}| + 1 \geq \exp\left[\frac{1 + \frac{pL}{2}}{k^2} \exp\left(\eta_t g_t \left(1 + \frac{pL}{2}\right)\right) + 1\right] \tag{9}$$

The previous line (9) is equivalent to:

$$k^2(\log(|w_{t+1}| + 1) - 1) \geq \left(1 + \frac{pL}{2}\right) \exp\left(\eta_t g_t \left(1 + \frac{pL}{2}\right)\right) \tag{10}$$

Notice that $\psi(w_{t+1}) = (|w_{t+1}| + 1)(\log(|w_{t+1}| + 1) - 1) + 1 \geq (|w_{t+1}| + 1)(\log(|w_{t+1}| + 1) - 1)$. Then multiplying (10) by $\eta_t|g_t|$ we have

$$(|w_{t+1}| + 1)\left(1 + \frac{pL}{2}\right)\exp\left[\eta_t|g_t|\left(1 + \frac{pL}{2}\right)\right]\eta_t|g_t| \leq k^2\eta_t|g_t|\psi(w_{t+1}) \tag{11}$$

Combining (8) and (11), we see that (9) implies

$$g_t \cdot (w_t - w_{t+1}) \leq k^2 \eta_t g_t^2 \psi(w_{t+1})$$

Now we bound $\left(\frac{1}{\eta_{t-1}} - \frac{1}{\eta_t}\right)\psi(w_{t+1})$:

$$\frac{1}{\eta_{t-1}} - \frac{1}{\eta_t} = k\sqrt{2}\left(\sqrt{M_{t-1} + \|g\|_{1:t-1}^2} - \sqrt{M_t + \|g\|_{1:t-1}^2 + g_t^2}\right)$$

$$\leq k\sqrt{2}\left(\left[\sqrt{M_t + \|g\|_{1:t-1}^2 + g_t^2} - \frac{g_t^2 + M_t - M_{t-1}}{2\sqrt{M_t + \|g\|_{1:t-1}^2 + g_t^2}}\right] - \sqrt{M_t + \|g\|_{1:t-1}^2 + g_t^2}\right)$$

$$\leq -k\sqrt{2}\frac{g_t^2}{2\sqrt{M_t + \|g\|_{1:t}^2}}$$

$$= -k^2\eta_t g_t^2$$

Thus when (9) holds we have

$$\left(\frac{1}{\eta_{t-1}} - \frac{1}{\eta_t}\right)\psi(w_{t+1}) + g_t(w_t - w_{t+1}) \leq -k^2\eta_t g_t^2\psi(w_{t+1}) + k^2\eta_t^2 g_t^2\psi(w_{t+1}) \leq 0$$

Therefore our objective is to show that our conditions on $w_t$ imply the condition (9) on $w_{t+1}$.

First, we bound $\eta_t g_t$ in terms of $|w_t|$. Notice that

$$|w_t| + 1 = \exp\left(\frac{|g_{1:t-1}|}{k\sqrt{2}\sqrt{M_{t-1} + \|g\|_{1:t-1}^2}}\right)$$

$$\leq \exp\left(\frac{\sqrt{p}\sqrt{|g_{1:t-1}|}}{k\sqrt{2}}\right)$$

$$\frac{2k^2\log^2(|w_t| + 1)}{p} \leq |g_{1:t-1}|$$

Using this we have:

$$\eta_t g_t = \frac{g_t}{k\sqrt{2}\sqrt{M_t + \|g\|_{1:t}^2}}$$

$$\leq \frac{g_t}{k\sqrt{2}\sqrt{M_{t-1} + \|g\|_{1:t-1}^2 + g_t^2}}$$

$$\leq \frac{g_t\sqrt{p}}{k\sqrt{2}\sqrt{|g_{1:t-1}| + pg_t^2}}$$

$$\leq \frac{L\sqrt{p}}{k\sqrt{2}\sqrt{\frac{2k^2}{p}\log^2(|w_t| + 1) + pL^2}}$$

so that we can conclude:

$$\eta_t g_t \leq \frac{Lp}{k\sqrt{2}\sqrt{2k^2 \log^2(|w_t| + 1) + p^2 L^2}} \tag{12}$$

Further, by Lemma 7 we have

$$\frac{|w_t| + 1}{|w_{t+1}| + 1} = \exp(\eta_{t-1}|g_{1:t-1}| - \eta_t|g_{1:t}|)$$

$$\leq \exp\left[\eta_t g_t \left(1 + \frac{pL}{2}\right)\right]$$

Therefore we have

$$|w_{t+1}| + 1 \geq (|w_t| + 1)\exp\left[-\eta_t g_t \left(1 + \frac{pL}{2}\right)\right] \tag{13}$$

From (13), we see that (9) is guaranteed if we have

$$|w_t| + 1 \geq \exp\left[\eta_t g_t \left(1 + \frac{pL}{2}\right)\right]\exp\left[\frac{1 + \frac{pL}{2}}{k^2}\exp\left(\eta_t g_t \left(1 + \frac{pL}{2}\right)\right) + 1\right] \tag{14}$$

If we use our expression (12) in (14), and assume $|w_t| \geq \exp(L^2)$, we see that there exists some constant $C$ depending on $p$ and $k$ such that the RHS of (14) is $O(\exp(L))$ and so (14) holds for sufficiently large $L$.

For $p = 2/L$, $k = \sqrt{2}$, and $w_t \geq 15$ we can verify (14) numerically by plugging in the bound (12).

For the case $k = \sqrt{2}$, $|w_t| \geq 4\exp(p^2 L^2)$, we notice that by using (12), we can write (14) entirely in terms of $pL$. Graphing both sides numerically as functions of $pL$ then allows us to verify the condition.

$\square$

We have one final lemma we need before we can start stating some real regret bounds. This lemma can be viewed as observing that $\psi(w)$ is roughly $\frac{1}{D}$ strongly-convex for $|w|$ not much bigger than $D$.

**Lemma 12.** *Suppose $p \leq 2/L$, $k = \sqrt{2}$, $\|w_t\| \leq D$ and $\|g_t\| \leq L$ Then $g_t(w_t - w_{t+1}) \leq 6(\max(D + 1, \exp(1/2)))g_t^2 \eta_t$.*

*Proof.* By Theorem 5 it suffices to consider $\text{sign}(g_t) = \pm\text{sign}(g_{1:t-1})$.

We show that $|w_t - w_{t+1}| \leq 6(\max(D + 1, \exp(1/2)))|g_t|\eta_t$ so that the result follows by multiplying by $|g_t|$.

From Lemma 7, we have $|\eta_{t-1}|g_{1:t-1}| - \eta_t|g_{1:t}|| \leq \eta_t|g_t|\left(1 + \frac{pL}{2}\right) \leq 2\eta_t|g_t|$. Further, note that $\eta_t|g_t| \leq \frac{1}{k\sqrt{2}} = \frac{1}{2}$. We consider two cases, either $\text{sign}(g_{1:t}) = \text{sign}(g_{1:t-1})$ or not.

**Case 1: $\text{sign}(g_{1:t}) = \text{sign}(g_{1:t-1})$:**

$$|w_t - w_{t+1}| = |\exp(\eta_{t-1}|g_{1:t-1}|) - \exp(\eta_t|g_{1:t}|)|$$

$$= (|w_t| + 1)|\exp(\eta_t|g_{1:t}| - \eta_{t-1}|g_{1:t-1}|) - 1|$$

$$\leq 2(D + 1)\eta_t|g_t|\exp(2\eta_t|g_t|)$$

$$\leq 2(D + 1)\eta_t|g_t|\exp\left(\frac{2}{k\sqrt{2}}\right)$$

$$\leq 6(D + 1)\eta_t|g_t|$$

**Case 2: $\text{sign}(g_{1:t}) \neq \text{sign}(g_{1:t-1})$:** In this case, we must have $|g_{1:t}| \leq |g_t|$. Let $X = \max(\eta_t|g_{1:t}|, \eta_{t-1}|g_{1:t-1}|)$. Then by triangle inequality we have

$$|w_t - w_{t+1}| \leq 2\max(|w_t|, |w_{t+1}|)$$

$$\leq 2(\exp(X) - 1)$$

$$\leq 2X\exp(X)$$

$$\leq 2(\max(|w_t|, |w_{t+1}|) + 1)X$$

Since $|\eta_{t-1}|g_{1:t-1}| - \eta_t|g_{1:t}|| \leq 2\eta_t g_t$, we have $X \leq 2\eta_t g_t + \eta_t|g_{1:t}| \leq 3\eta_t|g_t|$ so that we have

$$|w_t - w_{t+1}| \leq 6(\max(|w_t|, |w_{t+1}|) + 1)\eta_t|g_t|$$

Finally, we have $|w_{t+1}| + 1 = \exp(\eta_t|g_{1:t}|) \leq \exp(\eta_t|g_t|) \leq \exp(1/2)$, so that

$$|w_t - w_{t+1}| \leq 6\eta_t|g_t|(\max(|w_t|, |w_{t+1}|) + 1)$$
$$\leq 6\max(D+1, \exp(1/2))\eta_t|g_t|$$

$\square$

Now we are finally in a position to prove Lemma 3, which we re-state below:

**Lemma 3.** *Set $k = \sqrt{2}$. Suppose $\|g_t\| \leq L$ for $t < T$, $1/L \leq p \leq 2/L$, $g_T \leq L_{\max}$ and $L_{\max} \geq L$. Let $W_{\max} = \max_{t\in[1,T]} \|w_t\|$. Then the regret of FTRL with regularizers $\psi_t(w) = \psi(w)/\eta_t$ is:*

$$R_T(u) \leq \psi(u)/\eta_T + 96\sqrt{M_T + \|g\|_{1:T}^2} + 2L_{\max}\min\left[W_{\max}, 4\exp\left(4\frac{L_{\max}^2}{L^2}\right), \exp(\sqrt{T/2})\right]$$

$$\leq (2\psi(u) + 96)\sqrt{\sum_{t=1}^{T-1} L|g_t| + L_{\max}^2} + 8L_{\max}\min\left[\exp\left(\frac{4L_{\max}^2}{L^2}\right), \exp(\sqrt{T/2})\right]$$

$$\leq L_{\max}(2((\|u\| + 1)\log(\|u\| + 1) - \|u\|) + 96)\sqrt{T} + 8L_{\max}\min\left[e^{\frac{4L_{\max}^2}{L^2}}, e^{\sqrt{T/2}}\right]$$

*Proof of Lemma 3.* We combine Lemma 11 with Lemma 12: if $|w_t| \geq 15$ we have for all $t < T$:

$$\left(\frac{1}{\eta_{t-1}} - \frac{1}{\eta_t}\right)\psi(w_{t+1}) + g_t \cdot (w_t - w_{t+1}) < 0$$

and if $|w_t| \leq 15$ we have

$$\left(\frac{1}{\eta_{t-1}} - \frac{1}{\eta_t}\right)\psi(w_{t+1}) + g_t \cdot (w_t - w_{t+1}) \leq g_t \cdot (w_t - w_{t+1})$$
$$\leq 6 \times (15 + 1)\eta_t g_t^2$$
$$= 96\eta_t g_t^2$$

Therefore for all $t < T$ we have $\left(\frac{1}{\eta_{t-1}} - \frac{1}{\eta_t}\right)\psi(w_{t+1}) + g_t \cdot (w_t - w_{t+1}) \leq 96\eta_t g_t^2$.

$$R_T(u) \leq \psi(u)/\eta_T + \sum_{t=1}^{T}\left(\frac{1}{\eta_{t-1}} - \frac{1}{\eta_t}\right)\psi(w_{t+1}) + g_t \cdot (w_t - w_{t+1})$$

$$\leq \psi(u)/\eta_T + 96\sum_{t=1}^{T}\eta_t g_t^2 + \left(\frac{1}{\eta_{T-1}} - \frac{1}{\eta_T}\right)\psi(w_{T+1}) + g_T \cdot (w_T - w_{T+1})$$

We have

$$\left(\frac{1}{\eta_{T-1}} - \frac{1}{\eta_T}\right)\psi(w_{T+1}) < 0$$

so that

$$\left(\frac{1}{\eta_{T-1}} - \frac{1}{\eta_T}\right)\psi(w_{T+1}) + g_T \cdot (w_T - w_{T+1}) \leq 2L_{\max}W_{\max}$$

Further, again using Lemma 11 we have

$$\left(\frac{1}{\eta_{T-1}} - \frac{1}{\eta_T}\right)\psi(w_{T+1}) + g_T \cdot (w_T - w_{T+1}) < 0$$

for $|w_T| \geq 4\exp(p^2 L_{\max}^2)$ since $k = \sqrt{2}$.

Finally, notice that by definition of $\eta_t$ and $L$, we must have $|\eta_t g_{1:t}| \leq \frac{\sqrt{p|g_{1:t}|}}{k\sqrt{2}} \leq \sqrt{T/2}$, so that $\|w_t\| \leq \exp(\eta_t|g_{1:t}|) \leq \exp\left(\sqrt{T/2}\right)$. Thus we have

$$\left(\frac{1}{\eta_{T-1}} - \frac{1}{\eta_T}\right)\psi(w_{T+1}) + g_T \cdot (w_T - w_{T+1}) \leq 2L_{\max}\min(W_{\max}, 4\exp(4L_{\max}^2/L^2), \exp(\sqrt{2T}))$$

Now we make the following classic argument:

$$\sqrt{M_t + \|g\|_{1:t}^2} - \sqrt{M_{t-1} + \|g\|_{1:t-1}^2} \geq \frac{g_t^2 + M_t - M_{t-1}}{2\sqrt{M_t + \|g\|_{1:t}^2}}$$

$$\geq \frac{g_t^2}{2\sqrt{M_t + \|g\|_{1:t}^2}}$$

$$= \eta_t g_t^2$$

so that we can bound:

$$R_T(u) \leq \psi(u)/\eta_T + 96 \sum_{t=1}^{T} \eta_t g_t^2 + \left(\frac{1}{\eta_{T-1}} - \frac{1}{\eta_T}\right) \psi(w_{T+1}) + g_T \cdot (w_T - w_{T+1})$$

$$\leq \psi(u)/\eta_T + 96 \sqrt{M_T + \|g\|_{1:T}^2} + 2L_{\max} \min(W_{\max}, 4\exp(4L_{\max}^2/L^2), \exp(\sqrt{2T}))$$

To show the remaining two lines of the theorem, we prove by induction that $M_t + \|g\|_{1:t}^2 \leq L \sum_{t'=1}^{t} |g_{t'}|$ for all $t < T$. The statement is clearly true for $t = 1$. Suppose it holds for some $t$. Then notice that $|g_{1:t+1}| \leq |g_{t+1}| + |g_{1:t}|$. So we have

$$M_{t+1} + \|g\|_{1:t+1}^2 = \max\left(M_t + \|g\|_{1:t+1}^2, \frac{|g_{1:t+1}|}{p}\right)$$

$$\leq \max\left(M_t + \|g\|_{1:t}^2 + L|g_{t+1}|, L|g_{1:t+1}|\right)$$

$$\leq L \sum_{t'=1}^{t+1} |g_{t'}|$$

Finally, we observe that $M_T = \max\left(M_{T-1} + \|g\|_{1:T-1}^2 + g_T^2, \frac{|g_{1:T}|}{p}\right) \leq L_{\max}^2 + L \sum_{t=1}^{T-1} |g_{t'}|$ and the last two lines of the theorem follow immediately.

$\square$

## C   Additional Experimental Details

### C.1   Hyperparameter Optimization

For the linear classification tasks, we optimized hyperparameters in a two-step process. First, we tested every power of 10 from $10^{-5}$ to $10^2$. Second, if $\lambda$ was the best hyperparameter setting in step 1, we additionally tested $\beta\lambda$ for $\beta \in \{0.2, 0.4, 0.8, 2.0, 4.0, 6.0, 8.0\}$

For the neural network models, we optimized ADAM and ADAGRAD's learning rates by testing every power of 10 from $10^{-5}$ to $10^0$. For stochastic gradient descent, we used an exponentially decaying learning rate schedule specified in Tensorflow's (https://www.tensorflow.org/) MNIST and CIFAR-10 example code.

### C.2   Coordinate-wise updates

We proved all our results in arbitrarily many dimensions, leading to a dimension-independent regret bound. However, it is also possible to achieve dimension-dependent bounds by running an independent version of our algorithm on each coordinate. Formally, for OLO we have

$$R_T(u) = \sum_{t=1}^{T} g_t(w_t - u) = \sum_{i=1}^{d} \sum_{t=1}^{T} g_{t,i}(w_{t,i} - u_i) = \sum_{i=1}^{d} R_T^1(u_i)$$

where $R_T^1$ is the regret of a 1-dimensional instance of the algorithm. This reduction can yield substantially better regret bounds when the gradients $g_t$ are known to be sparse (but can be much worse when they are not). We use this coordinate-wise update strategy for our linear classification experiments for RESCALEDEXP. We also considered coordinate-wise updates and non-coordinate wise updates for the other algorithms, taking the best-performing of the two.

For all algorithms in the linear classification experiments, we found that the difference between coordinate-wise and non-coordinate wise updates was not very striking. However, for the neural network experiments we found RESCALEDEXP performed extremely poorly when using coordinate-wise updates, and performed extremely well with non-coordinate wise updates. We hypothesize that this is due to a combination of non-convexity of the model and frequent resets at different times for each coordinate.

| AdaGrad | RESCALEDEXP | AdaDelta | ScaleInvariant | Adam | PiSTOL |
|---------|-------------|----------|----------------|------|--------|
| 1.14 | 1.19 | 1.21 | 1.28 | 1.51 | 1.53 |

Table 1: Average normalized loss, using best hyperparameter setting for each algorithm.

## C.3 Re-centering RESCALEDEXP

For the non-convex neural network tasks we used a variant of RESCALEDEXP in which we re-center our FTRL algorithm at the beginning of each epoch. Formally, the pseudo-code is provided below:

---

**Algorithm 2** Re-centered RESCALEDEXP

---

**Initialize:** $k \leftarrow \sqrt{2}$, $M_0 \leftarrow 0$, $w_1 \leftarrow 0$, $t_\star \leftarrow 1$, $w_\star \leftarrow 0$
**for** $t = 1$ **to** $T$ **do**
    Play $w_t$, receive subgradient $g_t \in \partial \ell_t(w_t)$.
    **if** $t = 1$ **then**
        $L_1 \leftarrow \|g_1\|$
        $p \leftarrow 1/L_1$
    **end if**
    $M_t \leftarrow \max(M_{t-1}, \|g_{t_\star:t}\|/p - \|g\|^2_{t_\star:t})$.
    $\eta_t \leftarrow \frac{1}{k\sqrt{2(M_t + \|g\|^2_{t_\star:t})}}$
    $w_{t+1} \leftarrow w_\star + \text{argmin}_w \left[ \frac{\psi(w)}{\eta_t} + g_{t_\star:t} w \right] = w_\star - \frac{g_{t_\star:t}}{\|g_{t_\star:t}\|} \left[ \exp(\eta_t \|g_{t_\star:t}\|) - 1 \right]$
    **if** $\|g_t\| > 2L_t$ **then**
        $L_{t+1} \leftarrow \|g_t\|$
        $p \leftarrow 1/L_{t+1}$
        $t_\star \leftarrow t + 1$
        $M_t \leftarrow 0$
        $w_{t+1} \leftarrow 0$
        $w_\star \leftarrow w_{t-1}$
    **else**
        $L_{t+1} \leftarrow L_t$
    **end if**
**end for**

---

So long as $\|w_\star - u\| \leq \|u\|$, this algorithm maintains the same regret bound as the non-re-centered version of RESCALEDEXP. While it is intuitively reasonable to expect this to occur in a stochastic setting, an adversary can easily subvert this algorithm.

## C.4 Aggregating Studies

It is difficult to interpret the results of a study such as our linear classification experiments (see Section 4) in which no particular algorithm is always the "winner" for every dataset. In particular, consider the case of an analyst who wishes to run one of these algorithms on some new dataset, and doesn't have the either the resources or inclination to implement and tune each algorithm. Which should she choose? We suggest the following heuristic: pick the algorithm with the lowest loss averaged *across datasets*.

This heuristic is problematic because datasets in which all algorithms do very poorly will dominate the cross-dataset average. In order address this issue and compare losses across datasets properly, we compute a *normalized loss* for each algorithm and dataset. The normalized loss for an algorithm on a dataset is given by taking the loss experienced by the algorithm on its best hyperparameter setting on that dataset divided by the lowest loss observed by any algorithm and hyperparameter setting on that dataset. Thus a normalized loss of 1 on a dataset indicates that an algorithm outperformed all other algorithms on the dataset (at least for its best hyperparameter setting). We then average the normalized loss for each algorithm across datasets to obtain the scores for each algorithm (see Table 1).

These data indicate that while AdaGrad has a slight edge after tuning, RESCALEDEXP and AdaDelta do nearly equivalently well (4% and 6% worse performance, respectively). Therefore we suggest that if our intrepid analyst is willing to perform some hyperparameter tuning, then AdaGrad may be slightly better, but her choice doesn't matter too much. On the other hand, using RESCALEDEXP will allow her to skip any tuning step without compromising performance.