[Reviews · NeurIPS 2016]

Reviewer 1

Summary

The paper studies online convex optimization over a Hilbert space and without any assumption on the loss functions (besides convexity). This is a practically important optimization problem, as many machine learning problems can be reduced to it (via online to batch conversion). The paper does a nice job explaining the history of this problem. This paper is the first that lifts the assumption to know an a priori bound on the norms of the gradients of the loss functions and achieves an almost linear dependency on the norm of the competitor. (Previous papers either had suboptimal regret or assumed bounds on either the domain or the norm of the gradients.) The authors prove a lower bound on regret of any algorithm (Theorem 2.1). The lower bound shows that regret can be exponential in the worst case if the norms of the gradients grow too rapidly. The main result of the paper is an algorithm that roughly matches the lower bound. The algorithm is FTRL with regularizer (x+1)*log(x+1)-x and a clever learning rate schedule. Similar regularizers were considered before, however, the learning rate schedule is novel and very technical. The paper resolves a COLT 2016 open problem (Francesco Orabona and David Pal "Open Problem: Parameter-Free and Scale-Free Online Algorithms") by proving a lower bound. At the same it opens a host of new open questions, as it is clear that the proposed algorithm is not the final answer to the unconstrained online convex optimization problem -- but that's great news and most welcome! (E.g. doubling method in algorithm can be hopefully removed. Or one could imagine algorithms and regret bounds that are better under certain assumptions on the growth of the norms of the gradients.) As far as I can judge, the results are correct. I checked the lower bound in great detail. I checked the proof of the main theorem. However, I didn't check the lemmas in the appendix. ----- Suggestions for improvement: 1) Please add reference to COLT 2016 open problem by Francesco Orabona and David Pal, "Open Problem: Parameter-Free and Scale-Free Online Algorithms". It directly addresses the problem in the paper. You might get $100 award :) 2) The ratio |g_t|/L(t) is not defined for t=1. This is happens since L(1) is not defined. L(1) can be defined either as zero or minus infinity, but in the either case the ratio |g_t|/L(t) is not defined. More generally, the ratio will not be defined if first few g_t's are zero. Please explain what is meant. The largest confusion is in Theorem 2.1 where if the reader understands the ratio |g_1|/L(1) as plus infinity, the theorem does not hold. (Algorithm 1 is fine.) 3) In proof of Theorem 2.1. Please use gradients g_t instead of losses l_t(w) so that it matches statement of the Theorem (which uses only g_t's). I.e. please replace l_t(w)= -w with g_t = -1 and l_t(w)=2Tw with g_t=2T. 4) Also, in the proof of Theorem 2.1, between lines 83 and 84. The third inequality is really an equality. 5) Line 107: In the inequality (M_t + |g|_{1:t}^2 >= ...), M_t should be replaced by M_{t+1}. (Or shift indices on g_t's by one.) 6) Algorithm 1: A very minor issue arises when first few g_t's are zero. There are some divisions by zero in the algorithm. 7) Theorem 3.2. Please don't write g_T <= L_{max} >= L. I understand what was meant, but other readers might not. In particular, it is not clear if g_T < L < L_{max} is allowed. Simply, write g_T <= L_{max} and L_{max} >= L. 8) Please number the theorems simply as 1,2,3. Also, I would call Theorem 3.2 a lemma, since it's just technical result. 9) Though, experiments on neural networks are fashionable these days, the algorithm would shine on regression with absolute loss much better.

Qualitative Assessment

This is a good and honest paper on an important practical problem (stated as an COLT 2016 open problem). Research in this paper is a very big step towards a clean practical and optimal algorithm that will remove the need to "tune" learning rates for online convex unconstrained optimization problems.

Confidence in this Review

2-Confident (read it all; understood it all reasonably well)


Reviewer 2

Summary

This paper proposes a parameter-free algorithm in the unconstrained setting and claims to be optimal regret. The main contribution of this paper lies in two aspects: (1) It proves the ideal bound in the scenario where Lmax is known ahead is impossible to achieve when Lmax is unknown; (2) It proposes an algorithm by using guess-and-double strategy to adapt to the unknown Lmax. Experiments on both linear classification datasets and convolutional neural networks shows the algorithm is comparative when compared with five other algorithms.

Qualitative Assessment

The paper is well motivated and introduces a novel parameter-free algorithm, which offers both theoretical proof and experimental guarantees. It’s also well organized and the formalization is sufficiently explained. However, although this paper review several existing parameter-free algorithms in the very beginning, the authors show only limited comparison with these algorithms in the experiment section, which undermines the empirical results. Besides, the choice of p and k is rather arbitrary. Although it claims to achieve optimal regret, such an arbitrary choice leads to serious doubt on the optimality.

Confidence in this Review

2-Confident (read it all; understood it all reasonably well)


Reviewer 3

Summary

The authors introduce an algorithm for online optimization. The algorithm is on par with the state of the art, but is free of hyper parameters and does require additional tuning of the learning-rates.

Qualitative Assessment

The topic is interesting and there is distinct potential for impact. The results are well presented and the proofs seem sound. The numerical experiments look thorough, thought, the results may be analyzed a bit further.

Confidence in this Review

2-Confident (read it all; understood it all reasonably well)


Reviewer 4

Summary

In this paper, the authors propose an online convex optimization algorithm for the unconstrained setting without prior knowledge of any bounds on the loss functions. They first provide a lower bound to illustrate the difficulty of the learning problem, and then prove that the regret bound of the proposed algorithm is almost optimal.

Qualitative Assessment

The main contribution of this paper is the lower bound in Theorem 2.1 and the algorithm in Section 3, whose regret bound is given in Theorem 3.1. I have the following concerns/questions regarding these results. The lower bound in Theorem 2.1 is a bit weak. In my opinion, a better one should be stated as “For any $T$ and any $u$, there exists an adversarial strategy picking … such that …”. In contrast, the current theorem only holds for some $T$ and $u$. The three constants $c$, $k$, $\epsilon$ makes Theorem 2.1 confusing. For example, if we take $k=\|u\|^2$, it implies $\Omega(\|u\|^2)$ is unavoidable. If we increases the values of $c$ and $k$, the lower bound will approach infinity. How to interpret these behaviors? Although in Lines 49-54, the authors claim that a naïve application of “doubling trick” is insufficient, the algorithm in Section 3 is still based on the “doubling trick”. As there is an exponential term in theorem 3.1, the upper bound may be worse than that in [11]. Thus, the authors should provide a detailed comparison with [11]. In Section 4.1, the authors evaluate each algorithm with the average loss after one pass through the data. Why not plot the regret directly? To support the upper bound in Theorem 3.1, the authors should plot regret versus $T$, from which we can observe how the regret increases with $T$. From the experimental results, it seems AdaDelta is highly comparable to RESCALEDEXP. What is the main advantage of RESCALEDEXP over AdaDelta? ----Updating------ The discussions among reviewers resolve my concern on the lower bound.

Confidence in this Review

2-Confident (read it all; understood it all reasonably well)


Reviewer 5

Summary

This paper studies online linear optimization with unconstrained domains and losses, providing a lower bound that rules out the ideal regret obtained when a bound on the gradients is known, and also a doubling-trick type of algorithm that matches the lower bound in terms of the horizon T. Some experiments for both linear model and neural network are conducted to show the effectiveness of this parameter-free algorithm.

Qualitative Assessment

Parameter-free online learning algorithms are in general an interesting subject. Understanding the gap between learning with and without the knowledge of the problem parameters would be very helpful. However, I have to say that the lower bound proof in the paper is very obscure and hard to understand. Specifically, while the construction of the adversary is pretty clear and simple, the parameters such as Equation (1), (2) and the specific choice of u in Case 1 seem to come out of nowhere. I failed to see why these choices are special and why one can’t get some other bounds (such as the optimal one when knowing L_max) by picking these parameters differently. I hope the authors could provide some more explanation in the rebuttal, and in general when revising the paper. I would suggest leaving the parameters unspecified first, and only argue what kind of tradeoff one can get by picking different choices at the end. The intuition behind the proposed algorithm in also missing. At the beginning the authors argue that doubling-trick does not work for previous work since the bound-violating gradient will ruin the algorithm entirely. So how exactly does the proposed algorithm avoid this? After all the algorithm still falls into the standard FTRL framework. So how is the special choice of the regularizer different from previous work and how is this helping the whole thing? Please explain. Some detailed comments: 1. Line 44, R_t(u) should be R_T(u)? 2. Last line of Case 1 in the proof of Thm 2.1, how is that an equality? 3. In Algorithm 1, it makes more sense to check the violation first instead of computing w_{t+1} first (which is wasted if the violation actually happens). 4. In Theorem 3.2, expression g_T \leq L_max \geq L seems weird and non-standard. 5. What exactly are the hyperparameters tuned for other algorithms in the experiments? 6. What is the conclusion for the experiments in Sec 4.1? 7. Last paragraph of Sec 5, “its regret matches the lower-bound”, why?

Confidence in this Review

2-Confident (read it all; understood it all reasonably well)


Reviewer 6

Summary

Thi paper studies the classic (static) online convex optimization that a learner aims to minimize its regret with respect to a fixed strategy. The authors first show that the regret can be bounded by a linear function when the loss is convex. Then, they provide a lower bound for the linear regret. Finally, they propose the RESCALEDEXP method which is an algorithm that tries to achieve this lower bound. The main idea of the proposed method, which is a variation of the follow the regularized learner method, is to approximate the maximum gradient norm of the loss functions which is unknown. The regret analysis of the proposed method is also provided. Numerical experiments compare the performance of the proposed method with existing methods in the literature.

Qualitative Assessment

My main concern is the definition of the lower bound. The authors show that the regret is bounded by the regret associated with a linear function. Then, they provide a lower bound for the regret corresponding to the linear loss. The point is we can not conclude that the lower bound provided in Theorem 2.1 is a lower bound for the original regret. Thus, this lower bound is not meaningful. Although the lower bound is meaningless, I think the regret analysis for the proposed method is interesting. The authors are able to show that the proposed method has a regret of order \|u\|log(\|u\|)L_max \sqrt{T} without prior knowledge of the parameter L_max. This result is novel and interesting. Regarding the lower bound, I think the authors should either clarify that lower bound in Theorem 2.1 does not hold for the original regret defined on page 1, or they have to remove it. The first line of the expression in Theorem 2.1 is also confusing since R_T(u) is used for general regret on page 1 while in Theorem 2.1 it is associated with the regret corresponding to linear losses.

Confidence in this Review

2-Confident (read it all; understood it all reasonably well)